# Genetic and Phylogenetic Analysis of Feline Coronavirus in Guangxi Province of China from 2021 to 2024

**DOI:** 10.3390/vetsci11100455

**Published:** 2024-09-25

**Authors:** Kaichuang Shi, Mengyi He, Yuwen Shi, Feng Long, Yandi Shi, Yanwen Yin, Yi Pan, Zongqiang Li, Shuping Feng

**Affiliations:** 1School of Basic Medical Sciences, Youjiang Medical University for Nationalities, Baise 533000, China; panyiyi2004@163.com; 2College of Animal Science and Technology, Guangxi University, Nanning 530005, China; 15638814692@163.com (M.H.); shiyuwen2@126.com (Y.S.); shiyandi123@126.com (Y.S.); lizq20@163.com (Z.L.); 3Guangxi Center for Animal Disease Control and Prevention, Nanning 530001, China; longfeng1136@163.com (F.L.); yanwen0349@126.com (Y.Y.)

**Keywords:** feline coronavirus, S gene, M gene, N gene, genetic evolution, phylogenetic analysis, recombination

## Abstract

**Simple Summary:**

Feline coronavirus (FCoV), as one of the important pathogens of feline viral gastroenteritis, has been attracting great attention. In this study, a total of 1869 rectal and nasal swabs, feces, and ascites samples from Guangxi province in southern China were collected during 2021–2024 and a positivity rate of 17.66% (330/1869) for FCoV was found. The nucleotide and amino acid homologies of S, M, and N genes were 81.2–99.6% and 70.2–99.5%, 89.9–100% and 91.6–100%, and 90.1–100% and 91.5–100%, respectively. All 63 FCoV strains obtained in this study belonged to type I FCoV (FCoV-I). Recombinant signals were detected in the S gene of FCoV strains GXLZ03-2022 and GXLZ08-2022, and in the CCoV strain GD/2020/X9. The results indicate that FCoV is still prevalent in Guangxi province, and the prevalent FCoV strains show high genetic diversity and novel epidemic characteristics.

**Abstract:**

Feline coronavirus (FCoV), as one of the important pathogens of feline viral gastroenteritis, has been attracting great attention. A total of 1869 rectal and nasal swabs, feces, and ascites samples were collected from eight regions in Guangxi province during 2021–2024. The multiplex RT-qPCR established in our laboratory was used to test these samples for FCoV, and 17.66% (330/1869) of the samples were positive for FCoV. The S, M, and N genes of 63 FCoV-positive samples were amplified and sequenced, and the genetic and evolutionary characteristics were analyzed. Similarity analysis showed that the nucleotide and amino acid homologies of S, M, and N genes were 81.2–99.6% and 70.2–99.5%, 89.9–100% and 91.6–100%, and 90.1–100% and 91.5–100%, respectively. Phylogenetic analysis revealed that all 63 FCoV strains, based on S gene sequences, belonged to type I FCoV (FCoV-I), and were clustered with Chinese strains and the Netherlands UU strains. Recombinant signals were detected in the S gene of strains GXLZ03-2022, GXLZ08-2022, and CCoV GD/2020/X9. The results suggest that FCoV is still prevalent in the Guangxi province of southern China, and the prevalent FCoV strains show high genetic diversity and novel epidemic characteristics.

## 1. Introduction

Feline coronavirus (FCoV) belongs to the order Nidovirales, family Coronaviridae, genus *Alphacoronavirus* [1]. FCoV is an enveloped, unsegmented, single-stranded, positive-sense RNA virus with corona-like appearance under electron microscopy [2]. The full length of FCoV genome is about 29 kb and it has a typical genome structure of coronavirus. It contains 11 open reading frames (ORFs) and encodes four structural proteins (spike, membrane, envelope, and nucleocapsid proteins) and seven nonstructural proteins (NSP1a, NSP1b, NSP3a, NSP3b, NSP3c, NSP7a, NSP7b). Structural proteins play an important role in protecting the viral genome and facilitating the interaction of virions with susceptible cells [3]. The S protein is not only the standard for inter-lineage classification, but also an important epitope and evolutionary marker for viral invasion and immune evasion. Particularly, it plays a significant role in driving cross-species transmission and host adaptation [4,5]. Therefore, the S gene is usually taken as the focus for analysis of viral genes. In addition, M and N proteins also play vital roles in viral functions, and were also usually used as the targeted genes for epidemiological surveillance and evolutionary analysis of coronaviruses [6,7,8,9].

Based on clinical signs and pathological changes, FCoV can be categorized into two biotypes: feline enteric coronavirus (FECV) and feline infectious peritonitis virus (FIPV) [10]. FECV infection typically results in a mild intestinal disease or in no obvious clinical signs that may resolve spontaneously. Cats recovering from FECV infection may harbor a persistent viral infection. Conversely, FIPV can lead to fatal feline infectious peritonitis (FIP), with a mortality rate up to 100%. The disease is characterized by serositis, systemic inflammation, and granulomatous lesions. Clinically, FIP manifests as either the exudative or the nonexudative type, both of which are associated with severe systemic diseases and produce varying degrees of pleural or peritoneal effusion [11]. FCoV is divided into two serotypes, type I (FCoV-I) and type II (FCoV-II) based on the differences in neutralizing the antibody response and the S protein gene sequence [2,3]. FCoV-I is widely circulated worldwide, accounting for 80–95% of natural infections; FCoV-II is only sporadically detected, generated by the recombination of FCoV-I and canine coronavirus (CCoV) [12], which has been widely reported in Asia such as South Korea and Malaysia, with an infection rate as high as 33.3% [13,14]. FCoV-II exhibits favorable cell culture characteristics and could be successfully cultured in vitro, whereas the growth of FCoV-I is limited to tissue culture cells and poses challenges for isolation. It may be due to the difference in the S gene between the two serotype strains that makes it difficult to perform epidemiological investigation of FCoV [15].

The prevalence of FCoV infections in cats is remarkably high, with a serum antibody positive rate ranging from 20% to 60%. However, the progression to FIP occurs only in approximately 5% to 12% of the infected felines [16,17]. At present, many mutation theories have been proposed regarding the conversion from FECV into FIPV. Several studies have suggested that the mutation of the S gene may be associated with a heightened virulence and tissue tropism of the virus [18,19], and the presence of a truncated or deleted 3c gene is frequently linked to the FIPV biotype [20].

Genetic recombination serves as the primary driving force behind the cross-species transmission of coronaviruses [21]. Occasionally, the recombination of certain coronaviruses in animal hosts can lead to their transmission and infection in humans, resulting in the onset of diseases [22]. The recent detection of porcine deltacoronavirus (PDCoV) and cat–dog reassortant α coronaviruses in human patients suggests that more animals may serve as potential hosts or intermediate hosts [23,24], which is necessary to analyze the recombination phenomenon in animal coronaviruses.

The partial or complete S gene sequences of FCoV have been sequenced and analyzed in central, northeastern, and southwestern China, reporting the epidemiological status and genetic evolution diversity of FCoV in China and revealing the coexistence of FCoV-I and FCoV-II strains in cats [25,26,27]. To investigate the prevalence and evolution of FCoV in the Guangxi province of China, clinical samples were collected from eight regions during 2021–2024 and tested for FCoV using a multiplex RT-qPCR in our laboratory [28]. The FCoV-positive samples were selected to amplify, sequence, and analyze S, M, and N genes. The nucleotide and amino acid similarity, phylogenetic trees, genetic evolution rate, potential recombination, and positive selection pressure were assessed to gain insights into the epidemiology and evolution of FCoV in the Guangxi province.

## 2. Materials and Methods

### 2.1. Sample Collection

Between November 2021 and March 2024, 1869 samples were collected from 23 pet hospitals in the Nanning, Liuzhou, Guilin, Baise, Hechi, Beihai, Qinzhou, and Yulin regions of the Guangxi province in southern China. These samples included 1827 rectal and nasal swabs/fecal samples and 42 ascites samples obtained from both subclinical and diseased unvaccinated cats. The samples were delivered to our laboratory at a temperature of ≤4 °C within 6 h post collection, and total nucleic acids were extracted using the TaKaRa MiniBEST Virus RNA/DNA Extraction Kit (Dalian, China) for detection of FCoV.

### 2.2. Primers Design

The specific primers for amplification of FCoV S, M, and N genes were designed to target the conserved regions of these genes (Table 1). The reference strain available in the NCBI Genbank (https://www.ncbi.nlm.nih.gov/nucleotide/, accessed on 10 September 2021) was the HLJ/HRB/2016/10 strain (Accession number: KY566209.1). The S gene was amplified by five overlapping fragments. All primers were synthesized by Guangzhou IGE Biotechnology Co., Ltd. (Guangzhou, China).

### 2.3. Test for FCoV in Clinical Samples

The 1869 clinical samples were tested for FCoV using our previously developed multiplex RT-qPCR [28]. The total viral RNA was used as a template, and the reaction procedures were as follows: 42 °C 5 min, 95 °C 10 s, 40 cycles of 95 °C 5 s, 55 °C 30 s. The samples with Ct values ≤ 36 were considered positive. The FCoV-positive samples were selected based on collection regions, dates, and Ct values for the gene sequencing of S, M, and N genes.

### 2.4. S, M, and N Gene Amplification and Sequencing

Sixty-three FCoV-positive samples were selected to amplify S, M, and N genes using the primers in Table 1. The 50 μL amplification system included 25 μL of Premix Taq (TaKaRa, Dalian, China), 1 μL of Forward/Reverse primer (25 pmol/μL) each, 5 μL of viral cDNA as template, and 18 μL of nuclease-free distilled water. The reaction procedures for the M gene and S1–S5 fragments were as follows: 95 °C 3 min, 35 cycles of 94 °C 30 s, 58 °C 30 s, 72 °C 60 s, 72 °C 10 min. The reaction procedures for the N gene were as follows: 95 °C 3 min, 35 cycles of 94 °C 30 s, 56 °C 30 s, 72 °C 75 s, 72 °C 10 min.

The PCR products were identified through electrophoresis. The targeted fragments were purified using the MiniBEST Viral DNA/RNA Nucleic Acid Extraction Kit (TaKaRa, Dalian, China), connected to the pMD18-T vector (TaKaRa, Dalian, China) overnight at 16 °C, then transformed into *E. coli* DH5α competent cells (TaKaRa, Dalian, China) and incubated in SOC medium at 37 °C for 1 h. A total of 100 μL of bacterial solution was evenly coated on an agar plate for incubation overnight at 37 °C. The positive clones were cultured in LB medium solution at 37 °C for 22–24 h and sequenced by IGE Biotechnology Co., Ltd. (Guangzhou, China).

### 2.5. Sequence Analysis

All FCoV gene sequences used in this study were collected from the NCBI GenBank (https://www.ncbi.nlm.nih.gov/nucleotide/, available on 7 March 2024), including sequences from other countries and other provinces in China. A total of 71 complete genomes, 32 S, 22 M, and 24 N gene sequences, as well as the 63 S, 63 M, and 63 N gene sequences obtained in this study were analyzed (Appendix A). All the sequences underwent multiple sequence alignment using the MEGA X software (https://www.megasoftware.net/archived_version_active_download, accessed on 7 March 2024). Subsequently, the nucleotide (nt) and deduced amino acid (aa) homologies of S, M, and N genes were analyzed and compared using the Clustal W algorithm in the DNAstar 7.0 software (https://www.dnastar.com/software/, accessed on 7 March 2024).

### 2.6. Phylogenetic Analysis

To understand the evolutionary relationship between the Guangxi strains and other FCoV strains from different countries, the phylogenetic trees based on S, M, and N gene sequences were generated using the MEGA X software. The optimal nucleotide replacement models for the S, M, and N genes were calculated as GTR+G+I. The phylogenetic tree was inferred using the maximum likelihood (ML) method and each ML tree was tested 1000 times with a Bootstrap test to estimate the branch support.

### 2.7. Homologous Recombination Analysis

The recombination detection program (RDP) software (http://web.cbio.uct.ac.za/~darren/rdp.html, accessed on 7 March 2024) was used to analyze recombination events of S gene sequences. RDP5 encompasses an array of sophisticated algorithms designed to detect and analyze recombination, including seven fundamental detection programs (RDP method, GENECONV, BootScanning, MaxChi, Chimaera, 3Seq, and SiScan) [29]. This software enables the identification of potential recombinant sequences from a set of aligned nucleic acid sequences, and it is used to detect and determine the precise location of a putative recombination breakpoint. It is generally considered that at least three of the seven methods can detect recombination, and a *p*-value cutoff of 0.05 (*p* < 0.05) is considered to be a real recombination. The SimPlot 3.5.1 software (https://github.com/Stephane-S/Simplot_PlusPlus, accessed on 7 March 2024) was used to determine the location of the assumed restructuring breakpoint.

Gene recombination between different viruses in the same genus is the main driving force of cross-species transmission of coronaviruses. Considering that FCoV, canine coronavirus (CCoV), and transmissible gastroenteritis virus (TGEV) belong to the α coronavirus genus and possess recombination potential, the domestic and foreign classical strains of CCoV and TGEV (Appendix A) were also downloaded for comparison with the obtained S gene sequences.

### 2.8. System Dynamics Analysis

The evolutionary rates of the S, M, and N genes were estimated using the BEAST software (v1.10.4) (http://beast.community/, accessed on 7 March 2024). The effective population size over time was simulated using a coalescent Bayesian skyline prior tree topology based on a GTR+G+I nucleotide substitution model and uncorrelated relaxed clock type. Two independent chains with a length of 1 × 10^8^ were run until convergence was achieved. Tracer (v1.7) (https://github.com/beast-dev/tracer/releases/latest, accessed on 7 March 2024) was used to check convergence and merge, with a burn-in of 10% of the total chain length. All parameter estimates yielded an effective sample size greater than 200 (ESS > 200).

### 2.9. Selection Pressure and Site Mutation Analysis

To understand the adaptive evolution of viruses, forward amino acid selection in the FCoV S gene was investigated using Datamonkey (http://www.datamonkey.org/, accessed on 7 March 2024). The methods included single likelihood ancestry counting (SLAC), fixed effect likelihood (FEL), mixed effect model of evolution (MEME), and fast unconstrained Bayesian approximation (FUBAR) for inference of selection. If a codon was highlighted by at least three methods, it was considered to be in a state of selection.

The focus, in this study, was on comparing the changes in the S gene locus in the ascites samples collected from suspected cases of FIP and other types of samples, with the FIPV C1Je strain (GenBank accession number: DQ848678.1) as a reference. According to the mutation hypothesis [30], the mutation status of the site 23,531 was investigated and the predicted amino acid variations were documented.

## 3. Results

### 3.1. Detection and Sequencing Results

The 1869 clinical samples collected from 2021 to 2024 were tested for FCoV using a multiplex RT-qPCR [29], and the positivity rate was 17.66% (330/1869). The positivity rates in eight regions within the Guangxi province are showed in Figure 1. In particular, 42 ascites samples that were clinically suspected to be FIP-positive had a positivity rate of 47.62% (20/42) for FCoV.

Sixty-three FCoV-positive samples were selected to amplify S, M, and N genes. After purification, transformation, and sequencing, 63 S, 63 M, and 63 N gene sequences of 63 FCoV strains from Guangxi province were obtained in this study. Their uploaded accession numbers to NCBI GenBank were PP448795–PP448854 and PP464266–PP464268 for the S gene, PP444724–PP444783 and PP464260–PP464262 for the M gene, and PP448735–PP448794 and PP464263–PP464265 for the N gene (Appendix A).

### 3.2. Similarity Analysis

The sequence analysis revealed that the nucleotide similarities of S, M, and N genes among FCoV strains from the Guangxi province were 81.2–99.6%, 89.9–100%, and 90.1–100%, respectively. The amino acid similarities were 70.2–99.5%, 91.6–100%, and 91.5–100% for the S, M, and N genes, respectively. Compared with the reference sequences downloaded from the NCBI GenBank, the nucleotide and amino acid similarities were 68.9–90% and 58.1–82.8%, 79.8–95.9% and 80.8–98.5%, and 88.7–95.5% and 89.3–97.9%, respectively (Table 2). These indicate that the FCoV S gene exhibited relatively low similarity.

### 3.3. Phylogenetic Analysis

#### 3.3.1. Phylogenetic Analysis Based on S Gene Sequences

According to the S gene phylogenetic tree (Figure 2), all 63 strains obtained in this study were clustered into the type I genotype (FCoV-I), and no strain clustered into the type II genotype (FCoV-II), indicating that the FCoV-I strains were the dominant epidemic strains in the Guangxi province. Most strains from the Guangxi province were found to be in the same branch as the Chinese strains and the UU strains from the Netherlands, all of which had a close genetic relationship. Distinct evolutionary branches were observed in Australia, the United States, the United Kingdom, and Belgium, with notable geographical variations. The strains from the Taiwan province and other provinces of China were far apart in terms of evolutionary branches, showing certain genetic evolution differences. On the contrary, the strains from the Taiwan province and Japan were clustered together, which might be attributed to variations and geographical environments.

#### 3.3.2. Phylogenetic Analysis Based on M Gene Sequences

The M gene phylogenetic tree (Figure 3) revealed that the Guangxi strains had the same genetic characteristics as most FCoV strains in China and the Netherlands, and a few strains were closely related to those from the United States and Australia. In addition, sequences from the United Kingdom were clustered into one branch.

#### 3.3.3. Phylogenetic Analysis Based on N Gene Sequences

The N gene phylogenetic tree (Figure 4) revealed that the Guangxi strains were distributed in the same branch as most of the Netherlands strains and a few strains from the United States, indicating that the N gene was relatively conservative.

### 3.4. Genetic Evolution Rate

According to the results of the system dynamics analysis, the average genetic evolution rates of FCoV S, M, and N genes were estimated at 4.61 × 10^−4^ (95% HPD: 1.58 × 10^−4^–3.51 × 10^−3^), 4.85 × 10^−5^ (95% HPD: 3.20 × 10^−5^–1.03 × 10^−4^), and 3.07 × 10^−5^ (95% HPD: 1.81 × 10^−5^–7.15 × 10^−5^) substitution/site/year, respectively, indicating that the M and N genes exhibited relative conservation, while the S gene evolved rapidly and displayed substantial genetic variability.

### 3.5. Analysis of Homologous Recombination of the S Gene

The recombination analysis of S gene sequences of FCoV strains from Guangxi and reference strains (FCoV, CCoV, TGEV) from GenBank revealed the detection of three distinct recombination signals. These signals corresponded to the FCoV GXLZ03-2022 strain, GXLZ08-2022 strain, and CCoV GD/2020/X9 strain (Table 3). The first two recombination strains were generated by gene exchange between strains in different regions of the Guangxi province, and the third recombination signal occurred between FCoV and CCoV, with detailed breakpoint information presented in Figure 5. The observed recombinant strain of CCoV-IIa GD/2020/X9 (accession number: MZ320954.1) in the reference sequence demonstrates a high potential for recombination between coronaviruses within the same genus. It was primarily derived from the FCoV-II ZJU1617 strain (accession number: MT239439.1) from China, exhibiting a similarity of 97.3%. Additionally, it had a secondary parent from the CCoV-II TGEV-like CCoV 341/05 strain (accession number: EU856361.1) from Italy, with a similarity of 99.6%. The sequence analysis and comparison of the S gene of FCoV-I, FCoV-II, CCoV, and TGEV revealed no evidence of recombination between FCoV and TGEV.

### 3.6. Positive Selection Pressure and Comparison of Mutation Sites of the S Gene

Under the influence of natural selection, evolution progresses toward adaptation to the environment. Therefore, the selection pressure of the S protein was investigated. Seven mutations that were the result of positive selection pressure were identified, located at positions 275, 290, 374, 378, 830, 854, and 980 (Table 4).

After comparison, four of five ascites samples showed T at the site of 23,531 nt in the S2 gene (Figure 6), while the remaining ascites sample and other samples exhibited A, resulting in a change from methionine (M) in FECV to leucine (L) in FIPV. Although the alteration of this site is not entirely correlated with the transition from FECV to FIPV, it provides some guidance for distinguishing between the two biotypes and remains an active research hotspot. Mutations of the S gene or of other genes will lead to the emergence of highly virulent pathogenic FIPV, but the location of the virulence conversion is not accurate enough.

## 4. Discussion

The increasingly intimate relationship between humans and pets may heighten the risk of viral transmission across different hosts, thereby elevating the likelihood of human and animal infection. Several severe infectious diseases in humans, such as acquired immune deficiency syndrome (AIDS), Ebola fever, avian influenza, and severe acute respiratory syndrome (SARS), are caused by cross-species transmission of zoonotic RNA viruses [31]. Given that FCoV belongs to the category of RNA viruses which are easily mutated and highly infectious [32,33], and there is a lack of effective vaccines to prevent and control the disease, it is necessary to comprehend the prevalence of FCoV in the Guangxi province.

All the 1869 clinical samples were collected from unvaccinated pet cats. The positivity rate of FCoV in the Guangxi province was 17.66%, significantly lower than the reported prevalence in other regions of China, such as central China (46.6%), southwest China (80.35%), and northern China (74.6%) [25,27,34]. On the one hand, it may be related to the source of the samples, because the samples used in this study contained a large number of subclinical animals. On the other hand, it could also be linked to lifestyle habits, as pet owners have been increasingly prioritizing their pets’ health and environmental conditions, which have been experiencing significant improvements. FCoV-I was the predominant strain in the Guangxi province, a fact that is consistent with the prevalence in other regions of China [25,26,27,34,35]. According to the classification statistics, the positivity rate of FCoV in ascites samples was 47.62% (20/42), and the positivity rate of FCoV in rectal and nasal swabs/fecal samples was 16.97% (310/1827). The detection rate of FCoV in ascites samples was significantly higher than that associated with other sample types. A study has demonstrated that the proportion of FCoV RNA detected in tissue samples of cats with FIP lesions by RT-qPCR (95%, 43/45) was much higher than that in tissue samples of cats without FIP (22%, 9/41) [19]. Additionally, the relative copy number of FCoV was significantly higher in FIP samples [36], which aligns with the results obtained from this study. It is noteworthy that co-infection of FCoV-I and FCoV-II are common in clinical samples in China [34], with a prevalence rate as high as 36.7% (51/139) [27]. However, no positive sample of FCoV-II were obtained in this study.

Phylogenetic analyses showed that the S gene, as a genotype classification marker, could divide FCoV into two major branches, while the phylogenetic tree based on the M and N genes could not reveal this relationship. FCoV strains exhibited clustering based on their geographical origins, with Guangxi strains in the same branch as the Chinese strains and the Netherlands UU strain; the Taiwan (China) strain was closely related to the Japanese strain, and Japan was also the country with more FCoV-II strains [37]. Each country’s strains have a distinct evolutionary branch, with significant geographic differences [38]. In addition, the evolution rate of the S gene is very high, accompanied by a certain degree of positive selection pressure, making it a good candidate to further investigate the transmission mode of FCoV [39].

The changes at the 23,531 nucleotide site in the S gene were detected in all fecal and one ascites samples, with A as the conserved nucleotide. Four ascites samples had mutations at this locus, indicating that the S gene had a high degree of genetic diversity. The relatively low homology and recombination events of the S gene between the obtained strains and the reference strains also illustrate the complex genetic evolution characteristics of FCoV. At present, the mutation at the M1058L site is primarily associated with change in viral tissue tropism, which is related to the systemic spread of the disease [40]. In addition, recombination of the genome plays a pivotal role in the evolution of viruses [41]. Since recombination events and mutations are the main factors influencing the molecular evolution of RNA viruses, it is crucial to take into account coronavirus recombination information across various species backgrounds. Deletions, insertions, mutations, and other phenomena commonly occur within the S gene and other genes [25,42]. Accurately pinpointing single nucleotide differences to determine biotype conversion from FECV to FIPV is challenging [43,44]. The development of FIP is the result of the interaction between the highly toxic FCoV and the immune system of the infected cat individual, which is subject to the influence of multiple complex factors such as genes and environment [2,42,45]. There is currently no cure or reliable means of prevention for the highly virulent strain FIPV. Furthermore, in light of the mechanism known as antibody dependent enhancement (ADE), the activation of the humoral immune response may exacerbate disease progression [46,47], increasing the difficulty of prevention and control of FCoV.

## 5. Conclusions

In this study, a total of 1869 rectal and nasal swabs/feces and ascites samples from the Guangxi province during 2021–2024 showed a positivity rate of 17.66% (330/1869) for FCoV. The nucleotide and amino acid homologies of S, M, and N genes were 81.2–99.6% and 70.2–99.5%, 89.9–100% and 91.6–100%, 90.1–100% and 91.5–100%, respectively. All 63 FCoV strains obtained in this study belonged to FCoV-I. Recombinant signals were detected in the S gene of the FCoV strains GXLZ03-2022 and GXLZ08-2022 and of the CCoV strain GD/2020/X9. The results indicate that FCoV is still prevalent in the Guangxi province of southern China, and the prevalent FCoV strains show high genetic diversity and novel epidemic characteristics.

## Figures and Tables

**Figure 1 vetsci-11-00455-f001:**
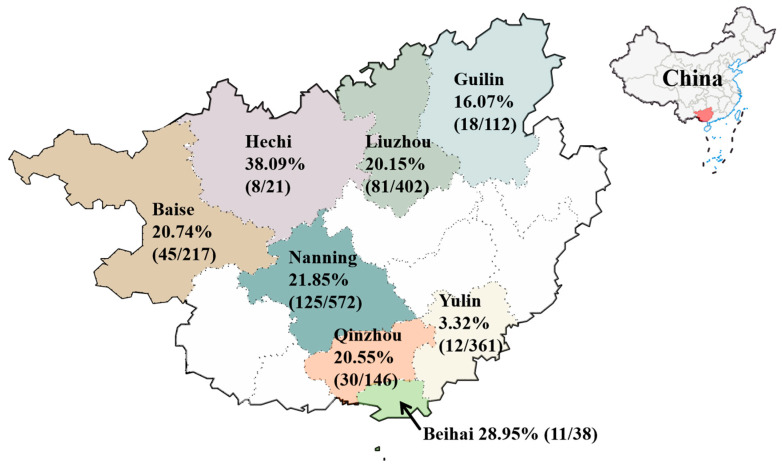
The positivity rates of FCoV in different regions of the Guangxi province, China.

**Figure 2 vetsci-11-00455-f002:**
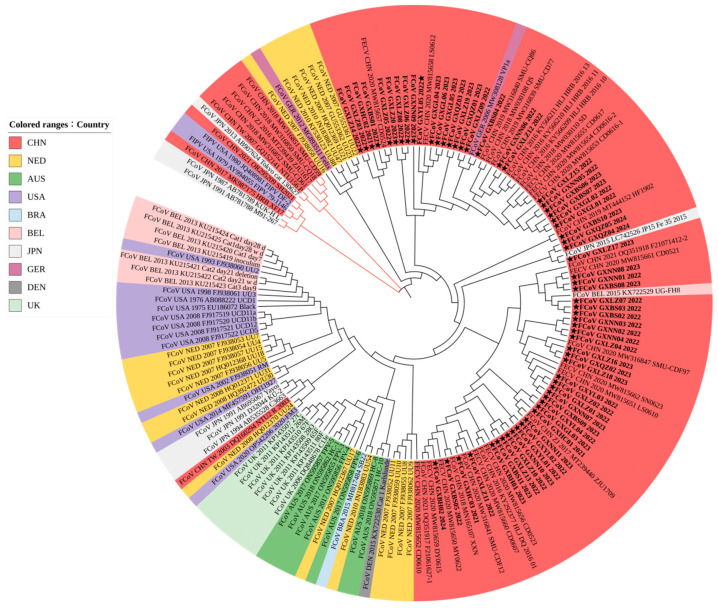
Phylogenetic analysis based on the FCoV S gene nucleotide sequences. The obtained 63 FCoV strains in this study are marked with a star (★). Black branches represent FCoV-I; red branches represent FCoV-II. The color of the label on the left indicates different countries.

**Figure 3 vetsci-11-00455-f003:**
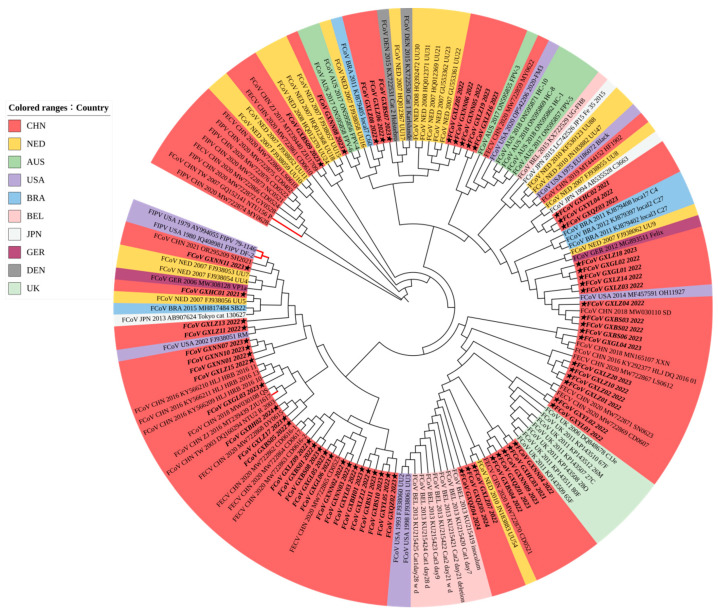
Phylogenetic analysis based on the FCoV M gene nucleotide sequences. The obtained 63 FCoV strains in this study are marked with a star (★). Black branches represent FCoV-I; red branches represent FCoV-II. The color of the label on the left indicates different countries.

**Figure 4 vetsci-11-00455-f004:**
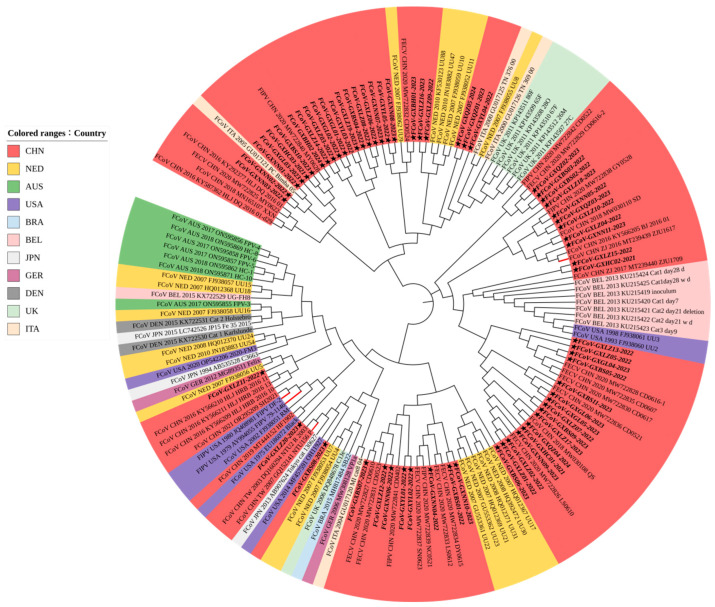
Phylogenetic analysis based on the FCoV N gene nucleotide sequences. The obtained 63 FCoV strains in this study are marked with a star (★). Black branches represent FCoV-I; red branches represent FCoV-II. The color of the label on the left indicates different countries.

**Figure 5 vetsci-11-00455-f005:**
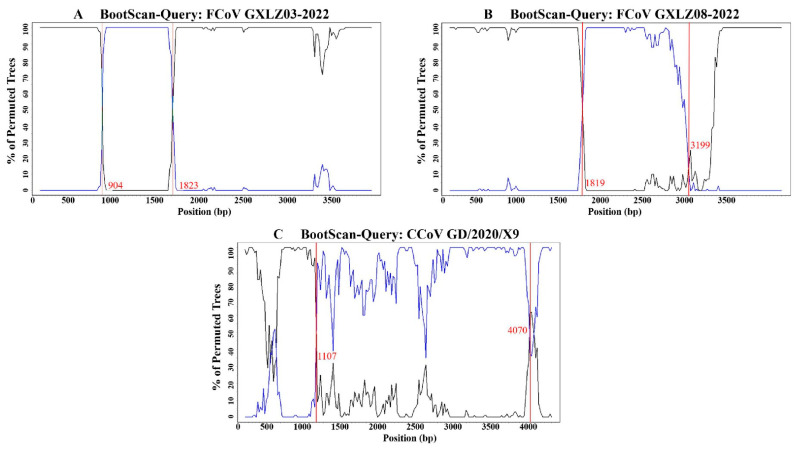
Recombination signals detected in SimPlot. Black represents the major parent, blue represents the minor parent, and the breakpoint positions are indicated by the red line.

**Figure 6 vetsci-11-00455-f006:**
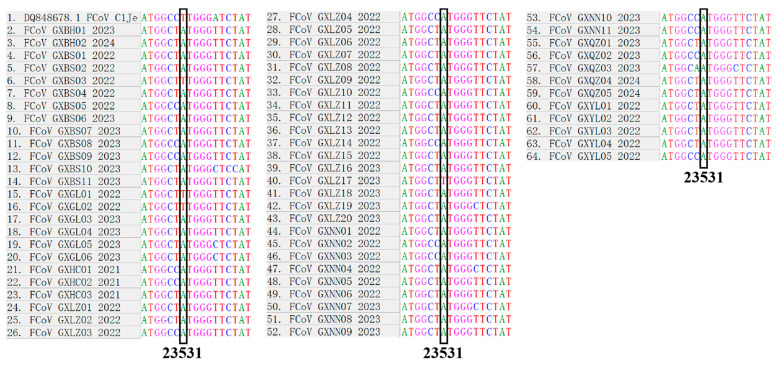
Changes at the 23,531 nucleotide site in the S gene of the FCoV strains from the Guangxi province.

**Table 1 vetsci-11-00455-t001:** The specific primers for S, M, and N gene amplification.

Primer	Sequence (5′→3′)	Position	Product/bp
FCoV-M-F	CCCGACGAAGCATTYTTGGTTTGAACTA	26,192–26,219	880
FCoV-M-R	GGAAGGTTCATCTCCCCAGTTGACG	27,047–27,071	
FCoV-N-F	GCACGTACTGAYAATTTGAGTGAAC	26,964–26,988	1202
FCoV-N-R	TGCGTTTAGTTCGTAACCTC	28,146–28,165	
FCoV-S1-F	GATATGGTYGTTGGATTRCTAAGG	20,376–20,399	960
FCoV-S1-R	TGCCRTCARTTGGCACAAAG	21,316–21,335	
FCoV-S2-F	GTGCCAATTGATGGCAAGATAC	21,320–21,341	918
FCoV-S2-R	TCCATTCGCCTGTGCTATTT	22,218–22,237	
FCoV-S3-F	ACTCTCACTTGCTGAYATACAC	22,192–22,213	982
FCoV-S3-R	TCTGTTACCATTGCAGACATACT	23,151–23,173	
FCoV-S4-F	AGGCCGAATACATTCAGATTCA	23,100–23,121	1032
FCoV-S4-R	TGCCACAGAAACCATACCTATC	24,110–24,131	
FCoV-S5-F	GTATGCTGAAGTCAAGGCTAGT	24,040–24,061	895
FCoV-S5-R	CAAGTACAGCGTCAACAGAGA	24,914–24,934	

**Table 2 vetsci-11-00455-t002:** Sequence similarity between obtained and reference sequences.

Gene	Sequence Similarity among Obtained Strains		Sequence Similarity between Obtained Strains and Reference Strains	
	nt	aa	nt	aa
S	81.2–99.6%	70.2–99.5%	68.9–90%	58.1–82.8%
M	89.9–100%	91.6–100%	79.8–95.9%	80.8–98.5%
N	90.1–100%	91.5–100%	88.7–95.5%	89.3–97.9%

**Table 3 vetsci-11-00455-t003:** The information on recombinant strains.

Recombinant Strain	Accession No.	Major Parent	Similarity	Minor Parent	Similarity
FCoV GXLZ03-2022	PP448818	FCoV GXLZ14-2022	99.2%	FCoV GXLZ05-2022	97.2%
FCoV GXLZ08-2022	PP448823	FCoV GXLZ06-2022	99.2%	FCoV GXBS01-2022	99.3%
CCoV GD/2020/X9	MZ320954	FCoV ZJU1617	97.3%	CCoV 341/05	99.6%

**Table 4 vetsci-11-00455-t004:** The analysis of positive selection pressure on the S protein of FCoV.

Site	FEL	SLAC	FUBAR	MEME
H275S	+	+	+	+
I290T	+	+	+	+
N374K	+	+	+	+
F378L	+	+	+	+
V830T	+	+	+	+
T854I	+	+	+	+
K980T	+	+	+	+

## Data Availability

Data are available within the article and Appendix A.

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
