# Peer review of "Genetic and Phylogenetic Analysis of Feline Coronavirus in Guangxi Province of China from 2021 to 2024"

_vetsci, 2024, doi:10.3390/vetsci11100455_

Round 1
Reviewer 1 Report
Comments and Suggestions for Authors
The manuscript is well presented, there are minor comments regarding terms used in the text, but nothing important.
Lane 11 and 21, Anal swabs or rectal swabs?
Lane 53 and others, change “symptoms” by “clinical signs”
Lane 57, change “latent” by “persistent”
Lane 104, change “asymptomatic” by “subclinical”
Author Response
The Cover Letter
September 16, 2024
Dear editor,
We have revised our manuscript carefully according to the reviewers’ suggestions. The point-to-point responses are as follows.
Reviewer #1
Comments and Suggestions for Authors
The manuscript is well presented, there are minor comments regarding terms used in the text, but nothing important.
- Lane 11 and 21, Anal swabs or rectal swabs?
Response: All “anal swabs” have been changed to “rectal swabs” in the revised manuscript. Please see Lines 12, 22, 104-105, 330, and 370 in the revised manuscript.
- Lane 53 and others, change “symptoms” by “clinical signs”
Response: All “symptoms” have been changed to “clinical signs” in the revised manuscript. Please see Lines 54, and 57 in the revised manuscript.
- Lane 57, change “latent” by “persistent”
Response: Change “latent” to “persistent” in the revised manuscript. Please see Line 58 in the revised manuscript.
- Lane 104, change “asymptomatic” by “subclinical”
Response: All “asymptomatic” have been changed to “subclinical” in the revised manuscript. Please see Line 105-106, and 324 in the revised manuscript.
Best regards,
Kaichuang Shi
Reviewer 2 Report
Comments and Suggestions for Authors
This study investigated feline coronavirus (FCoV) by analyzing 1,869 collected from Guangxi province between 2021 and 2024. A multiplex RT-qPCR method identified a 17.66% positivity rate for FCoV. DNA sequencing of the S, M, and N genes from 63 positive samples revealed high nucleotide and amino acid homology. Phylogenetic analysis classified all strains as type I FCoV, which clustered with other strain from China and Netherlands. Recombinant signals detected in specific strains indicate substantial genetic diversity and novel epidemiological characteristics of FCoV. The experiments are well-designed and the data are solid. And their findings will help us understand the prevalent of FCoV in South China. However, I have some comments:
1. Were the cats from which the samples were collected vaccinated? If so, which types of vaccines were administered? Is it possible that the sequences of the S, M, and N genes originate from the vaccinated virus? The vaccination status of cats in those areas should be discussed.
2. In coronaviruses, the S1 region of the spike gene exhibits considerable variation, whereas the S2 region remains relatively conserved. In Figure 6, in which subdomain do these mutations occur?
3. All abbreviations and acronyms used should be defined.
Author Response
The Cover Letter
September 16, 2024
Dear editor,
We have revised our manuscript carefully according to the reviewers’ suggestions. The point-to-point responses are as follows.
Reviewer #2
Comments and Suggestions for Authors
This study investigated feline coronavirus (FCoV) by analyzing 1,869 collected from Guangxi province between 2021 and 2024. A multiplex RT-qPCR method identified a 17.66% positivity rate for FCoV. DNA sequencing of the S, M, and N genes from 63 positive samples revealed high nucleotide and amino acid homology. Phylogenetic analysis classified all strains as type I FCoV, which clustered with other strain from China and Netherlands. Recombinant signals detected in specific strains indicate substantial genetic diversity and novel epidemiological characteristics of FCoV. The experiments are well-designed and the data are solid. And their findings will help us understand the prevalent of FCoV in South China. However, I have some comments:
- Were the cats from which the samples were collected vaccinated? If so, which types of vaccines were administered? Is it possible that the sequences of the S, M, and N genes originate from the vaccinated virus? The vaccination status of cats in those areas should be discussed.
Response: All the samples were collected from unvaccinated pet cats. This information has been described in the revised manuscript. Please see Lines 106, and 319 in the revised manuscript.
- In coronaviruses, the S1 region of the spike gene exhibits considerable variation, whereas the S2 region remains relatively conserved. In Figure 6, in which subdomain do these mutations occur?
Response: The mutation in Figure 6 was located at the site of 23531 nt in S2 gene. Please see Line 300 in the revised manuscript.
- All abbreviations and acronyms used should be defined.
Response: All abbreviations and acronyms used in this manuscript have been defined. Please see the revised manuscript.
Best regards,
Kaichuang Shi
Reviewer 3 Report
Comments and Suggestions for Authors
Different strains of Feline coronavirus (FCoV) infections cause diverse clinical outcomes in cats. Although there is no clear evidence to differentiate the feline enteric coronavirus and feline infectious peritonitis virus, the mutation in the spike protein gene might give some insight. Thus, the epidemiological and phylogenetic analyses of the circulant FCoV are essential. This study investigated the prevalence and phylogenetic characteristics of FCoV from the clinical samples in Guangxi province of China. Besides comparing the nucleotide and amino acid sequences, Shi et al., also conducted the genetic evolution rate, recombination event, and positive selection pressure analyses, and indicated the diverse gene characteristics of FCoV in the one specific area in China. This is a well-prepared manuscript that provides insight into the FCoV epidemic and the characteristics of the genome.
This paper mainly conducted an epidemiological and phylogenetic analysis of Feline coronavirus in Guangxi province, China. Shi et al., compared the nucleotide and amino acid sequences, as well as conducted genetic evolution rate, recombination event, and positive selection pressure analyses. The study indicated diverse gene characteristics of Feline coronavirus in this specific area of China.
This study utilized a sufficient number of clinical samples to analyze the sequences of Feline coronavirus circulating in the area, uncovering genome diversity and virus characteristics relevant to the special issue of the journal.
The topic of this study is original. However, although the surveillance study was limited to a restricted area, a sufficient number of samples collected compensates for this limitation. Additionally, analysis of homology and recombination events demonstrated that the Feline coronaviruses circulating in this area exhibit diverse characteristics and have undergone recombination events. As the spike gene may influence the clinical outcome of the Feline coronavirus, the recombination event and mutation identified in the spike gene during this study should be noted by other researchers and can provide valuable information for further analysis.
This manuscript is well-prepared and easy to read.
Author Response
The Cover Letter
September 16, 2024
Dear editor,
We have revised our manuscript carefully according to the reviewers’ suggestions. The point-to-point responses are as follows.
Reviewer #3
Comments and Suggestions for Authors
Different strains of Feline coronavirus (FCoV) infections cause diverse clinical outcomes in cats. Although there is no clear evidence to differentiate the feline enteric coronavirus and feline infectious peritonitis virus, the mutation in the spike protein gene might give some insight. Thus, the epidemiological and phylogenetic analyses of the circulant FCoV are essential. This study investigated the prevalence and phylogenetic characteristics of FCoV from the clinical samples in Guangxi province of China. Besides comparing the nucleotide and amino acid sequences, Shi et al., also conducted the genetic evolution rate, recombination event, and positive selection pressure analyses, and indicated the diverse gene characteristics of FCoV in the one specific area in China. This is a well-prepared manuscript that provides insight into the FCoV epidemic and the characteristics of the genome.
This paper mainly conducted an epidemiological and phylogenetic analysis of Feline coronavirus in Guangxi province, China. Shi et al., compared the nucleotide and amino acid sequences, as well as conducted genetic evolution rate, recombination event, and positive selection pressure analyses. The study indicated diverse gene characteristics of Feline coronavirus in this specific area of China.
This study utilized a sufficient number of clinical samples to analyze the sequences of Feline coronavirus circulating in the area, uncovering genome diversity and virus characteristics relevant to the special issue of the journal.
The topic of this study is original. However, although the surveillance study was limited to a restricted area, a sufficient number of samples collected compensates for this limitation. Additionally, analysis of homology and recombination events demonstrated that the Feline coronaviruses circulating in this area exhibit diverse characteristics and have undergone recombination events. As the spike gene may influence the clinical outcome of the Feline coronavirus, the recombination event and mutation identified in the spike gene during this study should be noted by other researchers and can provide valuable information for further analysis.
This manuscript is well-prepared and easy to read.
Response: Thanks very much for the reviewer's affirmation.
In addition, the manuscript has been revised carefully to decrease the total similarity to less than 30%, and the similarity to single paper to less than 5%. Please see the revised manuscript.
Best regards,
Kaichuang Shi
Reviewer 4 Report
Comments and Suggestions for Authors
Minor comment:
Under this section
3.6. Positive Selection Pressure and Comparison of Mutation Sites of S Gene
you have mentioned about seven amino acid mutations
Could you also make pymol image of those seven mutations in the pdb structure?
Author Response
The Cover Letter
September 16, 2024
Dear editor,
We have revised our manuscript carefully according to the reviewers’ suggestions. The point-to-point responses are as follows.
Reviewer #4
Comments and Suggestions for Authors
Minor comment:
Under this section
3.6. Positive Selection Pressure and Comparison of Mutation Sites of S Gene
you have mentioned about seven amino acid mutations
Could you also make pymol image of those seven mutations in the pdb structure?
Response: In our manuscript, we focus on the genetic and phylogenetic analysis of FCoV. Due to limited space, we did not make pymol image of the seven mutations in the pdb structure.
Best regards,
Kaichuang Shi
Round 2
Reviewer 1 Report
Comments and Suggestions for Authors
Thank you for your comments. In my opinion, the manuscript is suitable for acceptance.
Reviewer 2 Report
Comments and Suggestions for Authors
The authors have addressed all questions.
Reviewer 3 Report
Comments and Suggestions for Authors
This study primarily involved an epidemiological and phylogenetic analysis of the Feline coronavirus in Guangxi province, China, and indicated diverse gene characteristics of the Feline coronavirus in China. This revision adjusts certain descriptions and clarifies key points in the manuscript. The authors also made modifications to decrease the similarity to other published articles.
Reviewer 4 Report
Comments and Suggestions for Authors
Thanks for your response